# Interpretive Qualitative Evaluation Informs Research Participation and Advocacy Training Program for Seniors: A Pilot Study

**DOI:** 10.3390/healthcare11192679

**Published:** 2023-10-03

**Authors:** Allison A. Bay, Tina Tian, Madeleine E. Hackney, Hayley A. Silverstein, Ariel R. Hart, David Lazris, Molly M. Perkins

**Affiliations:** 1Department of Medicine, Emory University School of Medicine, 100 Woodruff Circle, Atlanta, GA 30322, USA; 2Center for Visual and Neurocognitive Rehabilitation, Atlanta Veterans Affairs Healthcare System, 1670 Clairmont Rd., Decatur, GA 30033, USA; 3Birmingham/Atlanta VA Geriatric Research, Education, and Clinical Center (GRECC), Decatur VA Research Building, 3101 Clairmont Rd., Mail Stop Code 11-B, Brookhaven, GA 30329, USA; 4Nell Hodgson Woodruff School of Nursing, Emory University, 1520 Clifton Rd., Atlanta, GA 30322, USA; 5Department of Rehabilitation Medicine, Emory University School of Medicine, 1441 Clifton Rd., Atlanta, GA 30322, USA

**Keywords:** barriers, facilitators, educational programs, clinical research, participation

## Abstract

***Background:*** An 8-week educational intervention co-taught by medical students and faculty was designed to foster communication between clinical researchers and populations of interest to ultimately increase participation in clinical research by older adults, including underrepresented groups. Weekly topics focused on age-related changes and health conditions, socio-contextual factors impacting aging populations, and wellness strategies. ***Objectives:*** To evaluate the successes and weaknesses of an educational intervention aimed at increasing the participation of older adults in clinical research. ***Design:*** A focus group was assembled after an 8-week educational intervention, titled DREAMS, to obtain participants’ feedback on the program, following a pre-formulated interview guide. ***Settings:*** Participants were interviewed in a health center office environment in the United States of America in April of 2016. ***Participants:*** A post-intervention focus group was conducted with a group of eight older adults (mean age = 75.8 ± 11.4 years) from 51 total participants who completed the intervention. ***Methods:*** The focus group was interviewed loosely following a pre-formed question guide. Participants were encouraged to give honest feedback. The conversation was recorded, transcribed verbatim, and analyzed using thematic analyses. ***Results:*** While participants viewed most aspects of the study as a success and stated that it was a productive learning experience, most participants had suggestions for improvements in the program content and implementation. Specifically, the composition of and direction to small breakout groups should be carefully considered and planned in this population, and attention should be paid to the delivery of sensitive topic such as death and dementia. A clear main benefit of this programmatic approach is the development of a rapport amongst participants and between participants and clinical researchers. ***Conclusions:*** The results provide useful insights regarding improving participation among hard-to-reach and historically underrepresented groups of older adults in clinical research. Future iterations of this program and similar educational interventions can use these findings to better achieve the programmatic objectives.

## 1. Introduction

The DREAMS (Developing a Research Participation Enhancement and Advocacy Training Program for Diverse Seniors) project is a two-part educational intervention targeting older adults to improve health literacy, to increase participation in future clinical trials from multiple racial, ethnic and socioeconomic backgrounds, and to encourage participants to be research advocates [1]. The DREAMS project sought to expand research participation in historically hard-to-reach populations, especially older adults, because limited diversity hinders progress in reducing health disparities [2], which is a national priority [3].

Older adults experience many age-related physiological changes that can result in increased difficulty performing activities of daily living, more falls, cognitive impairment, and other forms of functional deterioration. Many of these effects could be mitigated with increased awareness of physiological changes to better prepare older adults and their caregivers [4]. However, clinical investigators have not always sufficiently considered the importance of older adult participation in clinical trials [5]. Reasons for exclusion from research participation include ageism, a perceived lack of competence to participate, complications related to obtaining informed consent, and the need to allow more time for appointments to accommodate older individuals [6,7]. Barriers to the participation of older adults in research include transportation factors, unwillingness to undergo uncomfortable or risky procedures, poor vision, home responsibilities, mistrust of researchers, and low educational attainment [8,9]. Therefore, many of these barriers can be addressed through education and awareness of the purpose of clinical research, as well as through education of the researchers themselves about their biased perceptions of older research participants.

Similar to the barriers facing older adults, racial inequality in clinical trials is a well-documented phenomenon. The National Institutes of Health (NIH) Revitalization Act of 1993 requires NIH-funded clinical trials to include women and minorities as participants and assess outcomes by sex and race or ethnicity; however, targets for improving minority participation in research have not been met [10,11,12,13]. Nearly 40% of the population in the United States identifies as a racial or ethnic minority [14], but these racial and ethnic minorities are underrepresented in numerous domains of research, including cystic fibrosis [15], cancer [16,17], bipolar disorder [18], diabetes [19], cardiovascular disease [20], biobanking [21], and other research arenas, despite evidence that many illnesses and poorer health outcomes disproportionately burden underrepresented minority groups [22]. Focus groups with Black adults have suggested that fear and mistrust of the research and medical communities [23], lack of information, and knowledge of historical research-related aggrievances are major barriers to getting members of this community to volunteer as research participants [24]. Given the intersection of age and race, older, low-income adults of racial and ethnic minorities are particularly vulnerable to adverse health outcomes [2]. And with the fraught history of mistreatment of minority and disadvantaged groups by researchers [25], it is critical to have open dialogues with members of these groups to discuss the importance of research participation and to build participant trust in researchers and the clinical research process [26,27].

Focus groups conducted as part of a qualitative formative evaluation aimed at identifying potential barriers and facilitators before the implementation of DREAMS found that mistrust of the medical community and the belief that research is primarily conducted for profit as opposed to for the benefit of patients were major barriers [28]. This information was used to inform the implementation of the DREAMS project, which took place from 2015 to 2016 with four cohorts of older adults living in the southeastern United States whose sociodemographic makeup matched the locale [29,30,31]. This paper reports findings from a retrospective interpretive qualitative evaluation conducted post-DREAMS with eight DREAMS participants. This evaluation was designed to inform how future iterations of the DREAMS program could be enhanced through suggestions provided in a focus group format with the ultimate goal of improving the experiences of older adults throughout the clinical research process. 

## 2. Methods

This study was approved by the Emory University Institutional Review Board (IRB00080676). All participants provided written informed consent before participation.

### 2.1. Study Background: The DREAMS Program

The DREAMS project is an educational intervention developed to increase participation opportunities in all phases of the research process for older adults. DREAMS was inspired by the previous work of the authors, who had coordinated interactive lectures given by university faculty for older adults over several years [28,30,31,32,33,34,35]. Adults aged 55 years and older from across Metro Atlanta were recruited via several methods. Although chronologically, 65 years is considered to be the youngest age of “older adults”, in the southeastern United States, because of the high prevalence of comorbidities such as stroke, diabetes, and high blood pressure, many individuals experience sociological and biological aging before their chronological age [36,37,38,39]. Therefore, we enrolled participants who were 55 years of age or older. For the parent DREAMS project, the study team and partner organizations posted fliers, delivered presentations to stakeholders, and met with housing administrative staff of assisted living facilities who shared information with residents. Strong efforts were made to be inclusive of all the older citizens in the nearby metro area. Of the program participants, 41.3% were Black. Of note, the population of the city of Atlanta is 48.2% Black [40], and that of the the greater Atlanta metro area is 36% Black [41].

DREAMS consisted of weekly 90 min sessions over eight consecutive weeks and was delivered via interactive seminars focusing on health and wellness as well as educating participants about current translational and clinical aging research. The program was offered in four cohorts of approximately 20 to 25 participants who attended lectures together. Sessions were co-taught by researchers, faculty, and medical students (Table 1). Each class consisted of a 60 min lecture followed by a 30 min question and answer session. During the lecture, a medical student presented general information about the topic (e.g., macular degeneration) for the first 30 min. An expert or faculty member presented the second 30 min on current research information about the topic (e.g., research on therapies for macular degeneration). The last 30 min consisted of a moderated, small group discussion to encourage deeper processing of the learned information [31]. Participants were then broken up into 4 or 5 breakout groups to discuss the information presented in the lecture. The presenters and moderators each chose a group to join and were also encouraged to move amongst different groups to further discussions.

### 2.2. Focus Group Participants

Fifty-one participants completed DREAMS [28], which was defined as completing at least six of the eight lessons. Because of the need for focus groups to be intimate, a group size of 8 was ideal, and recruitment for the post-DREAMS focus group was ended after 8 participants had volunteered to participate (Figure 1). Sociodemographic information can be found in Table 2. The three men and five women who participated were an average of 75.8 ± 11.4 years old, had 3.0 ± 1.9 comorbidities, took 6.0 ± 5.3 medications, and mostly drove their own vehicle (75%). Six of the participants were White, one was Black, and one was multiracial. Six participants were retired. Participants were at moderate risk for loss of function in performing activities of daily living (ADL), indicated by composite physical function (CPF) scores at or below 18 (out of 24). All participants had at least some college education and were generally highly educated, with 17.5 ± 2.0 years of education. 

### 2.3. Focus Group Questions

The semi-structured guide administered to participants began with questions assessing their expectations regarding their participation in DREAMS. They were asked about their expected or unexpected experiences and/or outcomes (Appendix A: Interview Guide). We probed their thoughts on the process and design of DREAMS, the diversity of DREAMS and research in general, the presenters and medical students, and the employment of small group discussions (breakout groups) after lectures. We finished by asking participants for recommendations on future lecture topics and changes to the discussion group format, and we encouraged participants to share any other general thoughts they had on the program. Questions about strengths, weaknesses, and recommendations for future iterations of DREAMS were asked to assess the potential for translation into a larger public health program and to provide a basis for a future formative evaluation of expanded versions of DREAMS [42].

### 2.4. Data Analysis

The focus group conversation was recorded and transcribed verbatim. NVIVO 11 software was used to facilitate data analysis. Employing a thematic deductive and inductive approach to analyze data [43], in the first stage of analysis, we created a codebook of a priori codes based on the aims of the evaluation (e.g., to determine whether the program met expectations and identify areas for improvement). As analysis progressed, emergent themes were added to the codebook. Two primary coders independently coded the transcript. A secondary coder reviewed the coding of the primary coders and helped reconcile coding differences. These initial codes provided the foundation for final theme identification [44]. Each participant was assigned a unique alpha-numeric code: the letter “E” to designate their participation in the focus group followed by a number.

## 3. Results

We identified seven key themes: diverse learning expectations, mixed opinions regarding instructors and topics, community-building aspects, limitations of the small group format, uncomfortable topics, and reasons for participation. We outline these findings below with a simplified outline arranged by topic in Table 3 and a concept indicator model in Figure 2. 

### 3.1. Diverse Learning Expectations

All participants agreed that the program met their expectations. When participants were asked to think back to their expectations of DREAMS before they began the course, many participants recalled having had expectations that learning topics would mainly focus on topics related to aging, with four of the eight participants stating that aging was central to what they anticipated learning. Many responses focused on specific disease states, with the most common one being dementia. Several participants reported that they had initially been interested in topics that focused on how aging would affect themselves or their loved ones. Participant E4 stated, “[I] wanted to learn more about the process of aging and what life was like for me physically or mentally because my husband[‘s]…father [was] buried about a year earlier with dementia. And I wanted to know what to look for in myself or in my husband and different issues that may come about”. E7 said that she wanted to learn about dementia because she “lost a couple of friends lately. And both of them were a little bit younger than [herself] that have come down with Alzheimer’s”.

The second most common response (three out of eight) was that participants had no expectation other than learning something new or interesting. When asked to think about his initial expectations, E6 explained he “didn’t really have any preconceived ideas... [and] did not know what to expect”. Participant E8 answered that he “didn’t exactly know what to expect, but…thought it would be interesting, and it has been very interesting”.

### 3.2. Mixed Opinions Regarding Instructors and Topics

The quality of instructors, clinicians, clinical scientists, and medical students appealed to many of the participants. Participants stated how much they appreciated both the expertise of the professionals and the candor and intelligence of the medical students. Participant E7 stated, “It was unexpected to have like experts in the field. I thought maybe just anyone might be givin’ (sic) the talk.” E5 “thought it was great to be able to talk to the doctors directly along with the nurses and the researchers”.

When speaking about the medical students, participant E2 reported, “I have been, the last few years, a little despondent about what the new generation of doctors going to look like…But seeing the students that came, they were so bright and so cheerful…[T]hat gave me a lot of hope for the future”. Other participants agreed that the medical students were a joyful and positive part of their experience, with one stating, “I’ve absolutely adopted [a medical student]. I want to take him home with me”. 

While the instructors were a favorable part of their experience, the lessons themselves received mixed reactions from these participants. Many aspects of the presentations were viewed positively, such as the variety of topics that were covered and the diversity of the participant groups, which allowed for an engaging discussion on “age, culture, everything and… really added a lot, cause it’s so different for everyone” (E5). However, the variability of topics (Table 1) was viewed negatively by some participants because they felt that “some of [the sessions] were better and some of them are not so good” (E3). Five participants felt that the 90 min sessions were insufficient in terms of length of time. Specifically, in certain sessions, the presenters did not have time to cover all the material they had prepared. Participant E4 stated “Not getting to the end of the packet of each presenter was the only thing I could think of as a negative”. Three of the participants suggested that the sessions would have been better if they had been a full two hours long.

Participants also had many ideas for additional topics to include in the program, including emotional aging, anxiety, memory, technology, dental, hearing, and vision. However, participants admitted that it would be hard to make a course selection perfect for every participant.

### 3.3. Community-Building Aspects

The participants often stated that the program gave them a sense of purpose. They felt that the environment in the group-learning sessions was welcoming, and the presenters validated all their questions and encouraged future participation. E2 described presenters as being very responsive and “dignified our questions”. Specifically, the participants reported enjoying their experiences with the program and “look[ed] forward to coming to sessions because it (sic) was no stress involved” (E2). 

Additionally, participants felt they could gain new relationships through the program while learning about relevant, relatable topics that pertain to their age-group. E7 said that she “made some wonderful friends” throughout the program. While three participants (E2, E4 and E5) did not believe that they made friends in the program, they said they looked forward to the sessions every week and that there were “lots of different people to interact with” and “thought that was real interesting to see the different people who could converse about certain things that were happening to them then and now or had happened to them before, [which was] very profound, better than making a friend”.

Most participants proclaimed that they were excited to come every time and felt that the staff was “super friendly”. E5 appreciated that they were able to talk directly to people in the health professions. The participants agreed that the diversity of knowledge within the group and what was presented made the experience interactive, educational, and interesting, making them more curious about other topics relevant to their age-group and research that deals with older adults. Most participants also enjoyed their interactions with the medical students. However, participant E1 expressed that he would have preferred that experts were present for the entire session, including the small group discussions, instead of the medical student presenter because sometimes, he said, student presenters “couldn’t answer questions…even though they tried”.

Four of the eight participants stated that they would do the program again if they had the opportunity to do so and would recommend that their friends and colleagues participate in something similar. E8 proclaimed that he knew he learned a lot and enjoyed the experience and that he would “tell anyone who wanted to do it, ‘come on’”.

### 3.4. Limitations of the Small Group Format

Some participants felt that the small breakout groups held in the last half hour of the sessions were not very effective. Several focus group participants felt they were missing critical information if they were not placed in the small group with the expert presenter. Participant E6 felt that the small groups may have limited discussion, especially if some participants were shy and felt “pressured by them” to seem intelligent. E2 stated that in small groups, “some people talk a lot”, which also limited the amount that all participants could contribute to the discussions.

Another participant (E8) reported having a hard time hearing in some of the presentation sessions. E8 also did not like the small groups and would have preferred to have had outlines or pre-prepared notes for the breakout groups because he felt that the different small groups were not discussing the same topics. Three participants, E2, E4, and E5, also agreed that they would have preferred a question guide for small groups to provide more structure.

Given that hearing impairment is common amongst older adults [45], difficulty hearing the presenters was mentioned as a barrier to enjoying the sessions, although the planners of the program had been aware of such age-related issues. E8 said, “Hearing is one of the things older people have problems with”, in reference to the speakers. Participant E2 agreed and stated, “I couldn’t hear [the presenters]”.

### 3.5. Some Topics Were Uncomfortable: Aging and Death 

Five participants reported that some topics, along with the discussion about aging in general, caused anxiety. For example, participant E4 reported that a friend who had started the DREAMS program dropped out after the session on Alzheimer’s disease because her mother had passed away from Alzheimer’s. She said, “I’m not ready for that, because either I’m going to take care of my husband, or he’s going to take care of me”. Also related to the Alzheimer’s lecture, E7 stated, “I learned a lot…enough to scare me”. Participant E5 said, “[O]ne thing that comes to my mind is the emotional part of aging…realizing you’re getting there. And you know your friends are dying and you may be the last one alive...it can be very melancholy”.

### 3.6. Reasons for Participation: Recruitment

Although not central to the content of DREAMS, knowing which recruitment strategies were most successful was of interest. When the facilitator asked, “How did you find out about this study,” responses show that the most frequent way participants became involved in this DREAMS research study was through outreach conducted by the DREAMS research staff. Additionally, others saw a flyer promoting the study or were previously interested in the topic due to personal circumstances such as being interested in dementia because a loved one has dementia.

## 4. Discussion

The growth of patient and public involvement in research has been substantiated by a holistic approach from service providers. Recruitment often fails to take into account what participants hope to gain when entering medical studies [46]. This interpretive evaluation provides valuable insights into how to facilitate educational programs for older adults and meet expectations through addressing their needs and desires during the experience. Our work agrees with other studies on the usefulness of examining program implementation processes from the perspective of different individuals [47].

During this post-DREAMS focus group, several important barriers and facilitators for the participation of older adults in research, expectations for the DREAMS program, and suggestions for future educational interventions designed for older adult audiences were identified. 

### 4.1. Implementation Expectations 

Four out of the eight focus group participants expected to learn more information about aging, particularly information related to dementia before beginning the DREAMS program. Others (three out of eight) did not have specific expectations, and just anticipated learning something new and interesting. The participants’ responses underscore the importance of providing engaging and relevant information. Furthermore, although participants knew and expected aging-related topics, some participants, both in the larger group and in the focus group, found the material difficult to process emotionally, given that the subject matters brought up ideas of death and functional decline. 

Other studies evaluating the expectations of older adults in the healthcare sphere also emphasized the importance of the accessibility and trustworthiness of the presented information [48]. Although topics on aging may be initially difficult to process, it has been shown that older adults with lower expectations or knowledge regarding the aging process tend to have lower levels of physical activity, thus contributing to poorer overall health [49]. A higher sense of self-efficacy in older adults is associated with lower health risk and better health overall [50]; therefore, educational programs such as DREAMS may also contribute to better health outcomes in older adults. The use of a focus group gives participants the opportunity to challenge programs and for researchers to better tailor programs to the needs and expectations of older adults, an attribute that could contribute to participants’ sense of control over their own health [51].

### 4.2. Implementation Successes

Social factors, including engagement with presenters and other participants, enhanced participants’ experiences in DREAMS. Most participants found the clinical professionals and medical students who taught the lessons to be an engaging and a favorable part of their overall experience. Participants made connections with each other and with researchers. Seeing the other members of their course cohort was a pleasure for some and may have enhanced compliance with and attendance of the program. 

Importantly, social isolation is linked to worsened health outcomes [52]. In older adults, a lack of socialization is associated with cognitive decline [53]. Increasing social ties for older adults, even virtual ones, has protective effects against feelings of loneliness [54]. Additionally, socialization and social support have been found to be particularly important for program participation and interpersonal engagement [51]. 

### 4.3. Effectiveness of Topics 

The variety of lesson topics was perceived as both a positive and negative aspect of DREAMS. The diversity of topics contributed to lively and intriguing discussions. However, variability among topics resulted in a lack of interest among participants in some of the lessons, likely due to the lack of personal relevance. In this audience with mixed interests and backgrounds, although the program had an overall theme of aging, presenting topics that engaged all participants was a challenge and perhaps not one that can be easily addressed. On the other hand, it may be that the courses themselves and the information that was delivered were less important than the building of relationships and an emerging rapport between participants themselves as well as between participants and health professionals/researchers.

### 4.4. Maintenance and Future Iterations of DREAMS

A definitive area for improving future educational programs for older adults is to be more sensitive to age-related fears and stigma such as impending death, cognitive decline, and sensory dysfunction. Given that several participants found the discussion of aging to be emotional and/or stressful, more emphasis could be given to the positive aspects of aging, such as gaining experience, obtaining knowledge, and being a resource to younger generations to offset discussions about age-related diseases. While DREAMS made some headway in the direction of discussing fraught and challenging topics of aging, future work should involve a geropsychologist to learn how to best present information that is deemed sensitive to older adults. While it is important to educate older adults and possible future patient populations about pathological processes, maintaining independence and quality of life initiatives are of vital importance for a growing and thriving older adult population. Additionally, including information about ways to combat ageism and refute negative cultural views about older adults would have been helpful to include.

Another area for improvement is providing more guidelines for communication during program activities, particularly regarding the small group discussions, which were perceived to have been limited by both the backgrounds of the small groups’ leaders/moderators and the differing personality types present in each group. Specifically, those who were not loud or assertive were not able to contribute their ideas as much as they would have liked. Some potential solutions are to create a more structured and uniform discussion, including a question guide, to recruit more active moderators who are consistently present throughout the program, and to have the presenters rotate to each small group, spending equal time with all the groups. This may also be addressed with more presenters or longer sessions, as suggested by a few members of the focus group. Additionally, providing the presenters with microphones during their presentations may address the concerns of participants with hearing impairments. The presenters can also encourage participants at the beginning of each lecture to speak up if they have difficulty hearing to better titrate the volume and to also encourage discussion. Furthermore, time management on the part of the moderators and presenters may improve satisfaction with in-person educational programs. 

## 5. Limitations

The major limitation of this study is that only 25% of the focus group consisted of historically minoritized participants, compared to 50% of the larger study group, which decreases the generalizability of these findings. Future studies should include more individuals from underrepresented groups, older adults with a lower socioeconomic status, and individuals with fewer years of education. These groups are often difficult to reach and to recruit into studies yet would increase generalizability and the applicability of the findings to the inclusion of disadvantaged groups in research. The original study occurred in the large metropolitan area of Atlanta, which also limits the generalizability of these results to this region. 

Furthermore, because this focus group only consisted of 8 out of 51 participants who volunteered to participate out of the larger DREAMS cohort, the validity and reliability of these conclusions are limited. It was not directly discussed with the other DREAMS participants why participation in the focus group was declined; therefore, the reasons for the limited participation cannot be easily elucidated. Participation in this focus group may be limited by time constraints, how much the participants enjoyed the larger DREAMS project, and how open participants are about discussing their thoughts in a small group setting, amongst other factors. It is likely that individuals with fewer barriers would have attended this extra session, which we recognize as a limitation of this study.

Finally, there is limited concrete evidence to support that the participants of the larger DREAMS study experienced heightened biological age compared to chronological age aside from living in the southeastern United States. Therefore, generalizing the results of this study, which included adults aged 55 and older, to the older adult population may be inappropriate, since “older adults” is classically defined as those 65 and older. The findings of this study therefore may be appropriately generalizable to adults aged 55 and older. Furthermore, it is important to note that the subject matters explored by the DREAMS program and discussed by the focus group are relevant to the experiences of older adult populations.

## 6. Conclusions

Overall, the DREAMS educational intervention program received generally positive feedback from the focus group participants. Although the eight participants in the focus group encompass a very small sample of the original DREAMS study cohort, they provided valuable insights on areas were successful (educational, welcome environment, high-quality instructors, and good variation of topics) and areas that could be improved upon (more time with experts, greater facilitation of small group discussions, broader range of educational topics). Future iterations of the DREAMS program will be able to implement these findings to improve the educational experience of participants. Additionally, more initiatives to increase accessibility (such as providing transportation) may improve the diversity in focus groups in the future.

## 7. Future Directions

This paper provides important knowledge for future implementation of DREAMS. Future efforts to improve the participation of older adults should cultivate an environment for community building and personal growth. Additionally, future iterations of DREAMS would benefit from consultations with geropsychologists who can offer helpful advice on how to present more sensitive information to older adults and suggest future health-related educational topics. Through the positive and open attitude of clinicians, researchers, and medical professionals in training, meaningful opportunities for interaction between older individuals from diverse backgrounds can be created to improve the health outcomes of the aging population.

## Figures and Tables

**Figure 1 healthcare-11-02679-f001:**
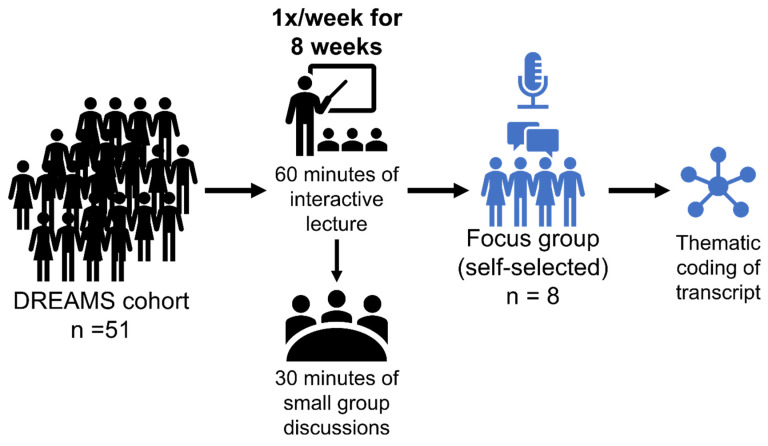
Summary of methods. A focus group consisting of 8 participants who volunteered out of a group of 51 participants of the DREAMS program, which consisted of 60 min interactive lectures once a week for 8 weeks followed by 30 min small group discussions, was created. The interview of the focus group members was recorded and transcribed. Afterwards, the transcription underwent thematic coding to elucidate common themes.

**Figure 2 healthcare-11-02679-f002:**
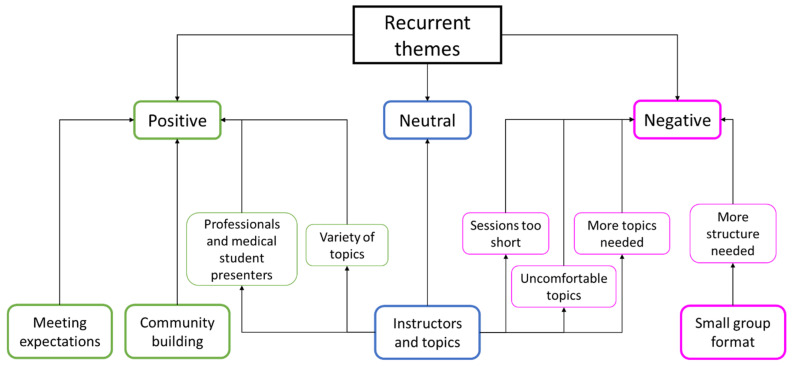
Concept indicator model of recurrent focus group themes. Positive themes are in green, neutral themes are in blue, and negative themes are in fuchsia.

**Table 1 healthcare-11-02679-t001:** List of topics presented during the DREAMS program by course. Participants could choose which course to take. Focus group participants by course: Course A = E1, E2, E4; Course B = E5, E6, E7; Course C = E8, Course D = E3.

Topics Presented in the DREAMS Program				
Session	Course A	Course B	Course C	Course D
Week 1	Research and Creativity in Later Life	Research and Creativity in Later Life	Research and Creativity in Later Life	Research and Creativity in Later Life
Week 2	Eyelid Ptosis and the Impairment of Vision	Bladder Matters in Aging Research	Role of Commensal Microbiota in Health Span	Role of Commensal Microbiota in Health Span
Week 3	End of Life, Palliative Care, Assisted Living	Dementia Family Caregiver Research	Tai Chi Studies: What have we Learned	Macular Degeneration- Fact or Fiction
Week 4	Hand Motor Function	Social Determinants of Health and Disparities	Neuromechanics Principles in Rehabilitation	Patient Perception of the Discharge Process
Week 5	Cardiovascular Health	Research in Specialized Nutrition Support	Cognition in Aging	Tai Chi Studies: What have we Learned
Week 6	Dementia Family Caregiver Research	Role of Commensal Microbiota in Health Span	Eye health	Cognition, Anesthesia and Older Adults
Week 7	Role of Commensal Microbiota in Health Span	Common Causes of Vision Loss	Cognition, Anesthesia and Older Adults	Pneumococcal Carriage Study in the Elderly
Week 8	Urinary Incontinence	End of Life, Palliative Care, Assisted Living	Balance and Falls in Individuals with Parkinson’s Disease	Balance and Falls in Individuals with Parkinson’s Disease

**Table 2 healthcare-11-02679-t002:** Descriptive summary of focus group participants (*n* = 8). All participants were healthy older adults. * Average excludes those who are not yet retired.

Post-DREAMS Focus Group Characteristics of Sample, *n* = 8	
Variable	Number (%) or *Mean (SD)*
**English First Language**	7 (87.5%)
**Race/Ethnicity**	
White	6 (75.0%)
Black	1 (12.5%)
Multiracial	1 (12.5%)
**Occupation Status**	
Full Time	1 (12.5%)
Part Time	1 (12.5%)
Homemaker	1 (12.5%)
Retired	5 (62.5%)
*Years Retired, M(SD) **	*22.2 (21.9)*
** *Education* **	*17.3 (2.4)*
**Use of Assistive Walking Device**	
Always	1 (12.5%)
Sometimes	3 (37.5%)
** *Age in Years* **	*75.8 (11.4)*
** *Number of Comorbidities* **	*3 (1.9)*
**Sex, M/F**	3M/5F (37.5% M/62.5% F)
**Marital Status**	
Single	1 (12.5%)
Married	3 (37.5%)
Separated/Divorced	1 (12.5%)
Widowed	3 (37.5%)
**Housing**	
House/Apt/Condo	3 (37.5%)
Independent Senior Housing	4 (50.0%)
Assisted Living	1 (12.5%)
**Transportation**	
Drive Own Vehicle	6 (75.0%)
Family/Friends Drive	1 (12.5%)
Transport Service	1 (12.5%)
** *Composite Physical Function/24* **	*17.6 (5.4)*
** *Number of Prescription Medications* **	*6 (5.3)*

**Table 3 healthcare-11-02679-t003:** Themes observed in focus group interview arranged by topic.

Recurrent Themes from Focus Group Interview	
Topic	Themes
Community Building	Welcome environment
	Expectations met
	Questions were valid
	Gave a sense of purpose
	Enjoyed coming to presentations
	Diverse study population
	Relatable/Friendly
Small group format does not work for everyone	Want more time with the presenter
	Small group may limit discussion, especially if shy
	Purpose of small group may not have been clear
	Possible added pressure on participants
Enjoyed both student and expert presenters	Students were friendly and caring, overall a positive benefit
	Quality instructors
	Would prefer an expert to always be present even if a student is teaching
Topics covered	Good variation of topics
	Learned a lot
	Longer sessions to learn more
	More topics related to old age including dentistry, arthritis, hearing, technology, eyes, etc.

## Data Availability

The datasets used and analyzed during the current study are available from the corresponding author upon reasonable request due to privacy restrictions.

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
