# Peer review of "Interpretive Qualitative Evaluation Informs Research Participation and Advocacy Training Program for Seniors: A Pilot Study"

_healthcare, 2023, doi:10.3390/healthcare11192679_

Round 1

Reviewer 1 Report (Previous Reviewer 1)

The authors are to be commended for their thorough and thoughtful revision of this manuscript.  

There were still some awkwardly constructed sentences and a few typographical errors in this revised version. It may be helpful to have someone other than one of the authors give a final proofreading. 

Author Response

Comments and Suggestions for Authors

The authors are to be commended for their thorough and thoughtful revision of this manuscript. 

RESPONSE: Thank you!

Comments on the Quality of English Language

There were still some awkwardly constructed sentences and a few typographical errors in this revised version. It may be helpful to have someone other than one of the authors give a final proofreading.

Thank you for your kind words. Further proofreading was performed with sentence structure adjustments. We believe we have removed the awkwardly constructed sentences.

Reviewer 2 Report (Previous Reviewer 3)

The authors have revised their paper with respect to previous feedback. Some additional comments:

***Table 2: Please present descriptive information as mean+-SD for continuous variables and frequency (%) for categorial variables. Only listing frequencies is incomplete, even with a low sample of n=8.

***Table 3: Bullet-points do not really belong in a table. Please revise.

***Line 227: Delete "(Table 1)". Representing tables disrupts prospective manuscript flow.

***Limitations: Including persons aged at 55-64 years and generalizing this age group to older adults is still not appropriate, despite your rationale provided. For example, the authors present no evidence that the sample has an accelerated biological age. This inappropriate age generalization needs to be acknowledged in the limitations. 

***Given the sample size and design, the title should end with "...Seniors: A Pilot Study".

Author Response

Comments and Suggestions for Authors

The authors have revised their paper with respect to previous feedback. Some additional comments:

***Table 2: Please present descriptive information as mean+-SD for continuous variables and frequency (%) for categorial variables. Only listing frequencies is incomplete, even with a low sample of n=8.

Thank you for this suggestion. Frequency percentages have been added to categorical variables in Table 2.

***Table 3: Bullet-points do not really belong in a table. Please revise.

We agree and have removed bullet point formatting in Table 3. This change will also prevent post-processing formatting that center the bullet point formatting, which likely would be distracting to readers.

***Line 227: Delete "(Table 1)". Representing tables disrupts prospective manuscript flow.

Deleted reference to “(Table 1)” as suggested.

***Limitations: Including persons aged at 55-64 years and generalizing this age group to older adults is still not appropriate, despite your rationale provided. For example, the authors present no evidence that the sample has an accelerated biological age. This inappropriate age generalization needs to be acknowledged in the limitations.

Thank you for this comment. Limitations regarding the age of participants is now discussed in the last paragraph of the “Limitations” section.

***Given the sample size and design, the title should end with "...Seniors: A Pilot Study".

Thank you for the suggestion. The title has been aptly adjusted.

Reviewer 3 Report (Previous Reviewer 4)

The authotrs implemented all my comments very well.

Author Response

Comments and Suggestions for Authors

The authors implemented all my comments very well.

Thank you for your kind words.

Reviewer 4 Report (New Reviewer)

The manuscript fulfils all the scientific standards necessary to be published.  The title explains the content of the manuscript well and the abstract includes necessary and sufficient data. Results and Discussion are sufficient and well-organized. The language of the manuscript is good enough to understand and there are not any spelling or punctuation mistakes. The manuscript has the sufficient quality and originality to be published in Healthcare.

Author Response

Comments and Suggestions for Authors

The manuscript fulfils all the scientific standards necessary to be published.  The title explains the content of the manuscript well and the abstract includes necessary and sufficient data. Results and Discussion are sufficient and well-organized. The language of the manuscript is good enough to understand and there are not any spelling or punctuation mistakes. The manuscript has the sufficient quality and originality to be published in Healthcare.

Thank you for your kind words.

This manuscript is a resubmission of an earlier submission. The following is a list of the peer review reports and author responses from that submission.

Round 1

Reviewer 1 Report

The results of a focus group interviewed after an 8-week educational intervention were presented. The objective was to review strengths and weaknesses of the educational intervention in order to inform future iterations of such an intervention, with a specific goal of increasing participation of underrepresented groups of older adults in research.

Introduction

Paragraph two: As written, this paragraph could be construed as very ageist. Please look again at what Jaul & Barron (2017) had to say, as what was written in your manuscript is a misrepresentation of their conclusions.

The entire paragraph needs to be rewritten and softened so as to acknowledge that age-related deficits in the domains you mentioned may occur, but are not a foregone conclusion in all older adults.

The last sentence in this paragraph does not make sense. 

Please update your references. 

Methods 

2.1 When speaking of diversity, you only talk about Black individuals. Were there any other aspects of diversity considered/noted? (I did not have access to Tables, Figures, or Supplemental materials.)

Paragraph 2: 2nd line is repeat of 1st line. 

2.2 Your focus group was not representative of your larger sample or of the area's population. 

You stated that "25% did not provide their own transportation, indicating some individuals present had diminished independence and/or disability." What evidence do you have for this statement? 

Results

4.3 Last paragraph: Is the Dillard et al. (2018) study the larger study? This is confusing. 

4.4 If 2 of the 8 members of the focus group could not hear the presenters, how valid are your results? 

Discussion

5.1 It would be helpful to discuss other literature on the influence of expectations on learning and compare and contrast with your findings. 

5.2 It would be helpful to discuss other literature on socialization and aging and compare and contrast with your findings. 

5.5 I would strongly suggest consulting with a geropsychologist regarding how to present sensitive information to older adults if this research is to continue, as your "positive aspects of aging" were weak. 

Some of the sentences were constructed awkwardly. It can be improved by a thorough proofreading. Reading the manuscript aloud can sometimes be helpful to detect awkward construction. 

Reviewer 2 Report

Title: Proper title

Problematic: Very interesting, it reveals concern in the evaluation of programs to promote literacy with elderly people, particularly those who are in a vulnerable situation, such as minorities. For this reason, it falls within the scope of the political priority given to this problem.

Abstract: Well-prepared, it integrates all parts of an abstract. However, it should be noted that the article is part of the Dreams Project.

The research design is in line with the objectives, and a qualitative and interpretative study is concerned with “information depth” and not “representativeness”. Thus, the focus group of eight seniors does not seem to me to be considered “representative”, for a number of reasons, taken up in the comment on the methods.

Methods: The design of the project is well structured and presented. If “This evaluation was designed to identify potentially needed modifications for future implementation of DREAMS and provide suggestions for im-proving the experience of diverse older adults throughout the clinical research process” (last paragraph of the introduction), I consider many limitations arising from the composition of the group focus, particularly in terms of race and education. The inclusion criteria that presided over the constitution of the focus group must be referred to in a clear and objective way. For these reasons and for methodological reasons as well, I consider that the term “representative” should be removed from everything related to the focus group.

Results: The results are adequately presented, in a global way. However, the phrases “This percentage contrasts with that the majority of participants from the larger sample that strongly agreed that they enjoyed the program or would like to continue participating (Dillard et al., 2018)” and “Given that hearing impairment is common amongst older adults (Ciorba, Bianchini, Pelucchi, & Pastore, 2012), difficulty hearing the presenters was mentioned as a barrier to enjoying the sessions, although the planners of the program had been aware of such age-related issues” respectively in points 4.3 and 4.4 must be included in the Discussion item.

Discussion: What is presented is an interpretation of results. The discussion would benefit from the articulation with the existing political orientations, with other studies and with the referenced bibliography. 

Limitations, Conclusions and future directions: very well achieved

References: Diverse bibliographic references, current and relevant to the subject under study. However, you need to correct/complete references number 15 and 25.

Reviewer 3 Report

HEALTHCARE-2570137 presents results for evaluation in research participation in older adults. While some parts of this work were interesting, other areas could use improvement, or were absent:

·         Introduction: “…participation in research have not been met. (Burchard…).” This sentence has two “.”. Please clean-up typos here and throughout.

·         Introduction: Last sentence, can you specify “diverse older adults”? Ethnically diverse? Age diverse?

·         Section 2.1: Please defend the 55+ age criteria for “older adults”. Persons aged 55-64 might be more so middle-aged.

·         Section 2.2: Much of this text can be reduced. Just use Table 2.

·         There are no figures or tables in the manuscript draft?!?

·         Make any changes to the abstract that align with those from the text.

Reviewer 4 Report

This paper presents the results of an 8-week educational intervention co-taught by medical students and fac[1]ulty that was designed to foster communication between clinical researchers and populations of interest to ultimately increase participation in clinical research by older adults, including underrepresented groups. Weekly topics focused on age-related changes and health conditions, socio-contextual factors impacting aging populations, and wellness strategies. The authos aim to Evaluate successes and weaknesses of an educational intervention aimed to increase participation of older adults in clinical research. They designed a focus group that assembled after an 8-week educational intervention to obtain their feedback on the program, roughly following a pre-formulated interview guide. Participants were interviewed in a health center office environment. PA post-intervention focus group was conducted with a representative group of eight older adults (mean age = 76 ± 11 years) from 51 total participants who completed the intervention. While participants viewed most aspects of the study as a success and stated that it was a productive learning experience, most participants had suggestions for improvements in the program content and implementation. Specifically, the composition of and direction to small breakout groups should be carefully considered and planned in this population, and attention should be paid to delivery of sensitive topic such as death and dementia. According to the authors a clear main benefit of this programmatic approach is the development of rapport amongst participants and between participants and clinical researchers.

This is an interesting paper from a societal and scientifical point of view.

I have the following points to make the paper stronger:

Abstract:

Mention the country and time period when the study has been conducted.

Introduction:

Mention the country and time period when the study has been conducted.

Add one or more research questions.

Method: only 8 out of 51 participants agreed to participate in the focusgroup. Explain why.

Discuss eventual consequences for validity and reliability.

Data analysis / Results

The authors come to seven key themes. Uae a concept indicator model to visualize your results and how the these are (not) interrelated.

Explain the letter E before the participant’s number.

Limitations: See my comments above related to Method.

Conclusion:

Answer one or more research questions.